# Steroid Metabolome Analysis in Dichorionic Diamniotic Twin Pregnancy

**DOI:** 10.3390/ijms25031591

**Published:** 2024-01-27

**Authors:** Andrej Černý, Martin Hill, Michala Vosátková, Zdeněk Laštůvka, Antonín Pařízek

**Affiliations:** 1Department of Gynaecology, Obstetrics and Neonatology, First Faculty of Medicine, Charles University and General University Hospital in Prague, 12808 Prague, Czech Republic; lastuvkaz@gmail.com (Z.L.); aparizek@seznam.cz (A.P.); 2Department of Steroids and Proteofactors, Institute of Endocrinology, 11000 Prague, Czech Republic; mhill@endo.cz (M.H.); mvosatkova@endo.cz (M.V.)

**Keywords:** foetomaternal steroidome, neuroactive steroids, multiple pregnancy, pregnancy complications

## Abstract

Steroid hormones have diverse roles in pregnancy; some help stabilise pregnancy and influence the stability of pregnancy and the onset of labour. Changes and disorders in steroidogenesis may be involved in several pregnancy pathologies. To date, only a few studies have performed a very limited steroid analysis in multiple pregnancies. Our teams investigated multiple pregnancies regarding the biosynthesis, transport, and effects of steroids. We recruited two groups of patients: pregnant women with multiple pregnancies as the study group, and a control singleton pregnancies group. Blood samples were drawn from the participants and analysed. Information about the mother, foetus, delivery, and newborn was extracted from medical records. The data were then analysed. The gestational age of twin pregnancies during delivery ranged from 35 + 3 to 39 + 3 weeks, while it was 38 + 1 to 41 + 1 weeks for the controls. Our findings provide answers to questions regarding the steroidome in multiple pregnancies. Results demonstrate differences in the steroidome between singleton and twin pregnancies. These were based on the presence of two placentae and two foetal adrenal glands, both with separate enzymatic activity. Since every newborn was delivered by caesarean section, analysis was not negatively influenced by changes in the steroid metabolome associated with the spontaneous onset of labour.

## 1. Introduction

Steroid hormones are synthesised in adrenal, gonadal, cerebral, and liver tissues. The placenta and the foetal adrenal zone (fetoplacental unit) are primarily responsible for the production of steroid hormones during pregnancy. A set of all steroid metabolites is called the “steroid metabolome”. Based on the mechanism of action, there are two kinds of steroids:Genomic: these bind to intracellular receptors or directly to DNA. This effect is slow; the response usually occurs hours or days later. They act as transcription factors that activate or suppress gene expression.Non-genomic: Their effect is associated with neurotransmitter receptors in the cytoplasmic membrane. These receptors affect ion channels and neuronal excitability. Changes based on this principle are considerably faster, ranging from seconds to milliseconds. Steroids with this effect are called “neuroactive steroids” or “neurosteroids”, and their mechanism of action was described between 1980 and 1990 [1,2].

Neurosteroids are steroid hormones synthesised in the central and peripheral nervous system. Neuroactive steroids are steroids that have neuroactive effects but can be synthesised in areas other than nervous tissue. Pregnane-type steroids, such as allopregnanolone, isopregnanolone, pregnanolone, and epipregnanolone, play an important role in pregnancy. Neuroactive steroids bind to several receptor types. GABAA-r (type A γ-aminobutyric acid receptor) and NDMA-r (N-methyl-D-aspartate receptor) are mostly affected. GABAA receptors influence chloride ion channels, and NDMA receptors influence calcium ion channels. These receptors are also located in the uterine muscle. This partially explains the effects of uterine contractility [3].

The endocrine activity of the foetoplacental unit during pregnancy is high. The adrenal gland and the foetal zone are the main organs for neuroactive steroid synthesis in the foetal body. Foetal zone activity is stimulated by the placental corticotropin-releasing hormone (CRH). The transport of hormones and their precursors within the foetoplacental unit is necessary due to the lack of enzymes in the placenta or foetus. At the end of the process, sufficient hormones for foetal development are produced [4].

### 1.1. Clinical Impact of Steroid Metabolome Studies

Several enzymes are involved in the synthesis and metabolism of steroid hormones. Defects in these systems can lead to complications during pregnancy. Neuroactive steroids are important for maintaining the stability of the foetoplacental unit. Changes in these levels are closely linked with the timing of delivery [5]. Previous research led by our team also compared the steroid metabolome between women who delivered spontaneously and those who delivered by caesarean section [6].

CRH is an important proteohormone involved in the timing of labour onset. Most of the CRH is produced in the placenta during pregnancy. Its influence on the foetoplacental unit appears to play a key role in the onset of labour. Some authors consider CRH to work as a “placental clock”. CRH levels are the highest at the beginning of delivery—almost ten thousand times higher than during pregnancy [7,8,9]. Right before the delivery, “relaxing” isoforms of receptors for CRH are converted to those that influence myometrium contractility. CRH also influences the effect of oxytocin and prostaglandin F2α. These processes explain the timing of delivery [10,11,12].

Hill’s study attempted to create a model for predicting labour onset. This study selected a group of neuroactive steroids that could be used as predictors of labour onset. Although the prediction models were better for umbilical blood, the model based on maternal blood was still very effective. In contrast to umbilical blood, using maternal blood for prediction is practically feasible and ethically acceptable [8].

The relationship between intrahepatic cholestasis during pregnancy (ICP) and steroid metabolome changes is well-known. In ICP cases, a lower production of adrenocorticotropic hormone (ACTH) and cortisone has been observed [13]. This may lead us to conclude that the CRH–ACTH axis may be altered. At the same time, the CRH–ACTH axis functioning is associated with the production of neuroactive steroids [14]. ICP in pregnancy has been studied by our team in several studies [15,16].

Pregnane steroids are involved in the regulation of blood pressure. Some authors reported significantly higher levels of allopregnanolone in women with chronic hypertension. In cases in which hypertension was complicated by preeclampsia, allopregnanolone levels were even higher. This study was based on a study from 1979 that investigated the relationship between levels of progesterone and 5α-dihydroprogesterone in patients with gestational hypertension. However, this relationship was not confirmed because the study was based only on the analysis of maternal serum and not umbilical cord blood. Progesterone levels in the amniotic fluid are significantly higher in women with later onset of preeclampsia [17,18]. This complication is caused by the dysregulation of angiogenic and anti-angiogenic growth factors in the placenta. Defects in placental oestrogen synthesis are believed to be one of the factors leading to preeclampsia [19].

Neuroactive steroids affect foetal growth, and alterations in the steroid metabolome may induce foetal growth restriction [20]. The effects of neuroactive steroids on the foetal and maternal central nervous systems are significant. The foetus’ central nervous system is in a state of permanent sleep, and foetal awareness is suppressed. Neuroactive steroids play an important role as neuroprotective agents. Allopregnanolone levels rise during acute hypoxia and protect the foetal brain during delivery. However, these mechanisms do not function during chronic hypoxia, and the foetus is thus at a higher threat [21,22,23].

Further, lower levels of neuroactive steroids are associated with a higher risk of depression during pregnancy or the postpartum period [24,25]. Coincidence between depression and allopregnanolone levels can be observed in the third trimester and later [26]. Pregnane steroids may alleviate pain perception during labour via negative modulation of T-type calcium channels [27].

Neuroactive steroids may also be used as therapeutic agents. Several projects have studied the benefits of neuroactive steroids for treating and preventing psychiatric diseases, such as depression, migraine, anxiety, panic attacks, and insomnia. The US Food and Drug Administration approved brexanolone (allopregnanolone) as an efficient anti-depression and anti-anxiety drug [28].

Neuroactive steroid therapy has been successfully used to treat nicotine and cocaine abuse [29]. Moreover, brexanolone has been investigated as part of the treatment for post-COVID acute respiratory distress syndrome [30]. In addition, ganaxolone (an allopregnanolone derivative) can potentially influence the neurological deficiency caused by premature labour [31].

### 1.2. Multiple Pregnancies

Multiple pregnancies are a reproductive health abnormality. This abnormality can however lead to serious obstetric complications. The population frequency was estimated using Hellin’s law. This rule approximates the frequency of twin pregnancy to 1:85, triplets to 1:85^2^, and quadruplets to 1:85^3^. Countries with higher frequency of assisted reproduction methods show higher frequencies of multiple pregnancies in their population.

Patients with multiple pregnancies have a significantly higher risk for serious pregnancy-related complications. Premature labour is one of the most threatening high-risk pathologies in multiple pregnancies [32,33]. Besides that, multiple pregnancy foetuses are in greater danger of intrauterine growth restriction in at least one of the foetuses, twin-to-twin transfusion syndrome, respiratory distress syndrome (RDS), and intrauterine stillbirth or abortion [34,35,36].

The endocrinology of multiple pregnancies has been the focus of several studies. The basis of these studies was the “twin testosterone transfer” hypothesis. This hypothesis states that androgen overproduction by the male foetus negatively affects the female foetus and induces masculine behavioural patterns, appearance, and cognition. This effect was also observed in mice. A Dutch experiment showed that maternal testosterone had no influence on female foetuses during pregnancy. Some authors believe that androgens from mothers are fully converted into oestrogens by the placenta [37]. However, other studies do not support or disconfirm this hypothesis [38,39]. In addition, foetuses in multiple pregnancies are not influenced by male siblings differently than when both foetuses are female [40,41].

In collaboration with the Department of Steroids and Proteofactors of the Institute of Endocrinology in Prague, our research also focused on investigating the biosynthesis, transport, and effects of steroids in multiple pregnancies. To date, no study has comprehensively analysed the steroidome in children and mothers with multiple pregnancies [42]. Therefore, our research aimed to clarify the relationships between foetuses and mothers, and between foetuses from the point of view of steroid synthesis and transport, as well as the physiology and pathophysiology of human pregnancy and childbirth. The study focused on differences based on arteriovenous differentiation.

### 1.3. Aims

The aims of the study of the steroid metabolome in dichorionic diamniotic twins were:

to describe details of the components of the steroid metabolome in twins,to show the difference between multiple and singleton pregnancies,to show the differences between male and female foetuses because the metabolism of the precursors is different.

Based on these aims and knowledge, we aimed to establish a model of steroidogenesis in multiple pregnancies in the same manner as that in singleton pregnancies.

## 2. Results

The list of metabolites analysed and their values for cord arterial blood and cord venous blood in singleton and twin pregnancies are given in Table 1 and for maternal venous blood in Table 2. Appendix A show the results of the OPLS models for distinguishing twin from singleton pregnancies based on steroid levels in umbilical arterial blood (Appendix A), umbilical venous blood (Appendix A), and maternal venous blood (Appendix A). Appendix A display statistically significant differences between female and male foetuses in terms of steroid levels in umbilical arterial blood (Appendix A), umbilical venous blood (Appendix A), and maternal venous blood (Appendix A). Values for the control group are included. After excluding samples that did not meet the criteria, we used 81 venous and arterial foetal blood samples, combined with 42 maternal blood samples. The control group included 47 foetal and maternal blood samples. The gestational age of twin pregnancies during delivery varied from 35 + 3 to 39 + 3 weeks. Controls were delivered at gestational ages of 38 + 1 and 41 + 1 weeks.

## 3. Discussion

Due to the major role of the placenta in the metabolism of steroids, we found it crucial to differentiate foetal blood from arterial and venous. Our research therefore analysed these blood samples separately. Comprehensive analysis in this large set of metabolites was never described in detail as we did.

The literature data regarding steroidogenesis and steroid transport in the fetoplacental unit relevant to our results are explained in detail in review articles [4,5]. Looking in more detail at the results obtained, we see that unconjugated C21 Δ^5^ steroids, and most of their unconjugated Δ^4^ and 5α/β-reduced metabolites, as well as some 5α/β-reduced C19 steroids and oestradiol, show higher levels in the umbilical vein and artery of twins and especially in the blood of their mothers (Table 1, Table 2 and Table 3, Appendix A).

This is to be expected, given that diamniotic dichorionic twins have two foetal adrenal glands and two placentas, with the foetal adrenal being the primary source of C21 Δ^5^ steroids in pregnancy, whereas it is the placental steroid sulphatase (STS) for C21 Δ^4^ steroids, which hydrolyses sulphates of C21 Δ^5^ steroids, mostly of foetal origin, to their unconjugated counterparts and then converts them to their Δ^4^ metabolites via HSD3B1, with progesterone being the main product of these processes. In addition, the placenta converts cholesterol and cholesterol sulphate (mostly of maternal origin) into pregnenolone, which is in turn metabolized into progesterone.

In addition, the placenta converts cholesterol and cholesterol sulphate (again predominantly of foetal origin) to pregnenolone, which is then metabolised again to progesterone and further in the placenta, foetal and maternal compartments to its C21 5α reduced metabolites, where the differences are most pronounced, and predominantly in the foetal liver to its C21 5β reduced metabolites, where the differences are less obvious as 5β-reductase activity is absent in the placenta in contrast to 5α-reductase.

Of the C21 ∆^5^ steroids, the most pronounced differences are for 16α-hydroxypregnenolone, where the very active 16α-hydroxylation enzyme CYP3A7 in the foetal liver is also involved.

The somewhat less pronounced differences in the 20α-dihydrometabolites of C21 steroids in umbilical cord blood are influenced by the tendency to convert them to 20-oxo analogues in the placenta–foetal direction, and the more pronounced differences in maternal blood by the opposite tendency in the placenta–mother direction. Both phenomena are explained in detail in our previous review [5].

In the case of testosterone levels, which do not differ significantly between singleton and twin pregnancies in all body fluids studied, our data are consistent with the study by Houghton et al. [42], and there are no differences in the cord blood levels of androstenedione either. For oestradiol, like the authors of this study, we also find higher levels in maternal serum, but in addition, we find this difference in umbilical cord venous blood. For estrone, estriol in maternal serum, we agree with the study by Kuiper et al. reporting higher levels of these steroids in maternal blood in twin pregnancies compared to singleton pregnancies [40]. For cord blood, our results and those of the previous study do not match.

The predominant absence of differences in umbilical cord blood for 17-hydroxymetabolites of C21 steroids may be related to the negligible CYP17A1 activity in the placenta compared with both foetus and mother so that the 2-placenta effect is lost here. However, in mothers of twins, the contributions from both foetuses are already additive and the differences here are mostly significant.

For many free androgens, oestrogens, 11-deoxycorticoids, 11β-hydroxy androgens, and corticosterone, there is again a clear trend towards higher levels in twins, and especially in their mothers. Exceptions are conjugated 11β-hydroxytestosterone with lower levels in umbilical arterial blood, cortisol, and cortisone, whose levels are lower in the blood of mothers of twins compared with mothers of singleton pregnancies.

Compared to unconjugated steroids, with a tendency towards higher levels in twins and especially in their mothers, conjugated steroids show the opposite trend, with lower levels in twins and, again, especially in their mothers. This phenomenon can be explained by the influence of two placentas and thus higher overall placental STS activity, leading to lower levels of conjugated steroids but a further increase in levels of their free analogues in twin pregnancies. This trend is very pronounced not only for the C21 ∆^5^ steroid pregnenolone sulphate but also for the C19 ∆^5^ steroid DHEA sulphate, and sometimes occurs with other steroid conjugates.

Finally, the results of the OPLS/OMR analysis show effective discrimination between singleton and twin pregnancies for arterial cord blood (sensitivity = 0.962 (0.888–1, shown as mean with 95% confidence interval), specificity = 1 (1–1)) (Appendix A), good discrimination for venous cord blood (sensitivity = 0.943 (0.866–1), specificity = 0.8 (0.598–1)) (Appendix A), and absolute discrimination for maternal blood (sensitivity = 1 (1–1), specificity = 1 (1–1)) (Appendix A). The results show that the overall doubling effect of foetal and placental steroidogenic tissues is most evident in maternal blood, although the levels of many steroids in maternal blood are significantly lower compared to cord blood.

Different levels of steroids (which are often bioactive) in singleton and twin pregnancies can influence a number of physiological and pathophysiological processes. Steroids act on a number of nuclear and ionotropic receptors, may positively or negatively influence the activity of detoxifying enzymes [43], and as a result may influence a number of pregnancy pathologies such as intrahepatic cholestasis of pregnancy [14], gestational diabetes [44], preeclampsia [17,18,19], foetal growth restriction [20], acute hypoxia [21,22,23], and postpartum depression [24,25]. The coincidence between depression and allopregnanolone levels can be observed in the third trimester and later [26], altered pain transmission and perception [27,43], altered gestational length [5], altered immune response [43,45], and possibly also programming of individual development in childhood and adulthood [46].

Regarding sex differences, our data show elevated levels of testosterone, 5α-dihydrotestosterone, and 5-androstene-3β,16α,17β-triol sulphate (which is a metabolite of one of the testosterone precursors androstenediol), with decreased levels of the other testosterone precursor androstenedione in arterial and venous umbilical cord blood (Appendix A), which may be related to steroidogenesis in the testes of male foetuses. Conversely, the interpretation of reduced levels of other steroids in the umbilical cord blood of male foetuses and significant differences in maternal venous blood remains (Appendix A) an open question for further research beyond the scope of this study. The literature reports the independence of key maternal blood steroids from foetal sex [37], and our data show differences for some steroids. However, interpretation of these results also requires further research beyond the scope of this study.

In discussing the differences between analytes in different body fluids, we found less obvious differences between arterial and venous blood. However, there were mostly very substantial differences between venous and maternal blood, which is consistent with our previous studies.

The steroid metabolome is closely linked to the timing of labour and affects several pregnancy pathologies. These components are well described in singleton pregnancies. A large set of steroid metabolome components, such as these, are within the multiple pregnancies described for the first time.

Because of the different pathways of steroid hormone synthesis, we differentiated between male and female foetuses and found some differences between the sexes that can be partly explained by testicular steroidogenesis in male foetuses. Some negative correlations of steroid levels with gestational age may be related to planning for a caesarean section. Most surgeries are performed before the calculated due date. The same results were observed in the control group (47 foetuses).

Because we excluded women with any issues that could have affected the results, our results showed no association with pregnancy pathologies. This should be the focus of a following study. Throughout the study, babies were delivered exclusively by planned caesarean section. The placental “timing” function had been eliminated. Therefore, our results did not show any changes in the steroidome related to the onset of labour. In addition, gestational age was included in the model as one of the covariates.

Our study may have several limitations we had to work with. Variation of gestational age is influenced by the date the pregnancy was terminated. This is managed by national guidelines for twin pregnancy care. In some cases of smaller foetuses, we were limited by the sample size—umbilical cord vessels are significantly thinner. Blood in these vessels clots almost immediately after cutting the umbilical cord. For proper analysis, at least 3 mL of blood is needed. In case of insufficient blood draw, analysis is almost impossible.

Investigation of dichorionic diamniotic twins raises a genetic issue; there may be cases in which embryos split in a very early phase, resulting in monozygotic twins with identical genetic information. Thus, the role of genetic influences, particularly in terms of enzymatic activity, can only be discussed theoretically.

Research and statistical analysis show that there are significant differences between singleton and dichorionic diamniotic twin pregnancies. Differences are based on the fact that there are two foetuses. Compared with singleton pregnancies, the foetal contribution can be considered as doubled in twins. Higher enzymatic activity is caused by two adrenal glands and two separate placentae.

This study demonstrates the scope for further research on multiple pregnancies. The next step should be the analysis of the steroid metabolome in multiple pregnancies with spontaneous onset of labour, in pathologies such as preeclampsia, or when signs of spontaneous premature labour are present. Further analysis based on this study can help to understand the pathophysiology of the pathologies mentioned above.

## 4. Methods and Materials

### 4.1. Study Participants

This study included two groups of patients: pregnant women with multiple pregnancies as the study group, and singleton pregnancies as the control group. Patient enrolment was conducted from November 2019 to November 2021.

All patients aged 18 years and above who met the inclusion criteria were asked to participate in the study. The eligibility criteria were twin pregnancy (for the study group), planned delivery to the author’s obstetrics clinic, and planned caesarean section due to multiple pregnancies. Dichorionic diamniotic pregnancies were terminated in the 34th–39th week of pregnancy.

We excluded women with metabolic disorders that could influence steroid metabolism, especially diabetes mellitus and intrahepatic cholestasis during pregnancy, women with intake of progesterone during pregnancy, women with acute malignant process, women with serious pathologies affecting pregnancy (e.g., twin-to-twin transfusion syndrome), women who were administered corticosteroids due to foetal lung maturation, smokers, and those who did not consent to participation in a study. We did not exclude IVF-conceived pregnancies.

The control group comprised women with singleton pregnancies. These patients were scheduled for caesarean section due to other medical conditions (e.g., caesarean section in the medical history, another type of uterine scar, breech presentation). The exclusion criteria were the same as for twin pregnancies (e.g., acute malignancy, serious metabolic disorder, disagreement).

Both groups included women after negative results of combined first-trimester screening and second-trimester morphology scan.

Blood samples were collected from 50 women with twin pregnancies and 20 with singleton pregnancies in the control group.

### 4.2. Specimen Collection and Pre-Laboratory Processing

Every consenting patient had venous maternal blood drawn at the latest possible time before the caesarean section and before administering any medication. Immediately after birth, arterial and venous blood were collected separately from each twin’s umbilical cord. We used the part of the cord attached to the placenta; therefore, the newborn was unaffected. For purposes of the research, we used the BDVacutainer^®^ K2EDTA Vacutainer system (Becton Dickinson, Franklin Lakes, NJ, USA). Blood volume was 6 mL from the mother and 3 mL from the umbilical artery and veins.

Samples were processed no longer than 2 h after collection. After clotting at room temperature, the sample was centrifuged for 10 min at 2400 rpm. The serum was then transferred to a microtube and frozen at −18 °C. The frozen samples were transported to the laboratory in an icebox. Several studies in the past showed that samples taken and frozen using this technique are very stable, and the concentrations of steroids do not change over time [47,48].

The same procedure was used for both the study group and the control group.

### 4.3. Laboratory Processing

All laboratory measurements were performed at the Department of Steroids and Proteofactors, Institute of Endocrinology, Prague, Czech Republic. Gas chromatography-tandem mass spectrometry (GC-MS/MS) was used for the sample analysis. The equipment included a gas chromatograph with automatic flow control, an autosampler, and a triple quadrupole detector with an adjustable electron voltage. The measurement is based on ionisation after electron impact. Hill developed, described, and published the method. The list of analysed metabolites is based on this study (see Appendix A) [49].

### 4.4. Clinical Data

Information about the mother, foetus, delivery, and newborn was extracted from medical records. The basic characteristic is provided in Table 3.

The information included the medical history of the pregnant women, age, education, race, general medical history, information about smoking and alcohol/drugs abuse, weight before and after pregnancy, height, parity, method of conception, biochemical and ultrasound markers of the foetus throughout the pregnancy, information about any problems during pregnancy, biochemical and blood test results from pregnancy (if provided by a gynaecologist providing regular checks during pregnancy), medication taken during pregnancy, vitals obtained during caesarean section, foetal sex, weight and length, APGAR score, pH and Astrup values, and information about postpartal adaptation.

### 4.5. Data Evaluation

In the first step, the power transformation parameters were found for each metric variable so that its distribution was as close as possible to the Gaussian distribution. The differences between twin and singleton pregnancies for each steroid were evaluated using a linear model consisting of factors Pregnancy type (Twin vs. Singleton) and Gender (Male vs. Female), adjusted for maternal age and gestational age at labour. The statistical software Statgraphics Centurion v. XVIII from Statgraphics Technologies, Inc. (The Plains, VA, USA) was used for the above analyses.

In addition, the differences between twin and singleton pregnancies (parameter status, twin vs. singleton pregnancy) were simultaneously tested for all steroids, maternal age and gestational age at labour in maternal, umbilical arterial, and umbilical venous blood, using multivariate regression models with a reduction in dimensionality known as orthogonal predictions to latent structure (OPLS) and ordinary multiple regression (OMR) [14,43].

Flow chart of OPLS analysis is added as Appendix A. Power calculation for the study is included in Appendix A.

The OPLS model, which is a multivariate regression with dimensionality reduction, allows for the evaluation of relationships between explained variables and the explaining variables (predictors) that may be highly correlated, which is also the case for steroids in metabolic pathways. The presence of the twin pregnancy in the OPLS model is expressed as the logarithm of the likelihood ratio (the ratio of the probability of the presence of twin pregnancy *p* to the probability of singleton pregnancy (1 *− p*)), i.e., the logarithm of the likelihood ratio is calculated, which then ranges from -infinity to + infinity. This approach ensures that the prediction of the probability of the presence of pathology is between 0 and 1 (after applying a recurrent formula that converts the prediction of the logarithm of likelihood ratio into a prediction of the probability of the presence of pathology).

The variability in the predictors is divided into two independent components. The first contains the variability of predictors that were shared with the probability of twin pregnancy (predictive component), whereas the orthogonal components explained the variability shared within highly correlated predictors. OPLS identifies relevant predictors, as well as the best linear combination of predictors to estimate the probability of the presence of pathology. After standardization of the variables, the OPLS model can be expressed as follows:(1)$$X=T_{p}P_{p}^{T}+T_{0}P_{0}^{T}+E$$
(2)$$Y=T_{p}P_{p}^{T}+F$$
where X is the matrix with predictors and subjects; Y is the vector of dependent variable and subjects; T_p_ is the vector of component scores from the single predictive component and subjects extracted from Y; T_o_ is the vector of component scores from the single orthogonal component and subjects extracted from X; P_p_ is the vector of component loadings for the predictive component extracted from Y; P_o_ is the vector of component loadings for the orthogonal component extracted from X and independent variables; and E and F are the error terms.

The relevant predictors were chosen using variable importance (VIP) statistics. The statistical software SIMCA-P v.12.0 from Umetrics AB (Umeå, Sweden), which was used for OPLS analysis, enabled the finding of the number of relevant components, the detection of multivariate non-homogeneities, and testing the multivariate normal distribution and homoscedasticity (constant variance).

The algorithm for obtaining the predictions was as follows:Transformation of the original data to obtain the values with symmetric distribution and constant varianceChecking the data homogeneity in predictors using Hotelling’s statistics and the eventual elimination of non-homogeneitiesTesting the relevance of predictors using variable importance statistics and the elimination of irrelevant predictorsCalculating component loadings for individual variables to evaluate their correlations with the predictive componentCalculating regression coefficients for the multiple regression model to evaluate the mutual independence of predictors after comparison with the corresponding component loadings from the OPLS modelCalculating predicted values of the logarithm of the ratio of the probability of twin pregnancy presence to the probability of singleton pregnancy (LLR)Calculating the probability of the twin pregnancy presence for individual subjectsCalculating the sensitivity and specificity of the prediction

## Figures and Tables

**Table 1 ijms-25-01591-t001:** Steroid differences between dichorionic diamniotic twin pregnancy and singleton pregnancy in umbilical venous blood.

	Umbilical Artery				Umbilical Vein			
	Pregnancy Type				Pregnancy Type			
Steroid	Singleton	Twin	*p*	η_p_^2^	Singleton	Twin	*p*	η_p_^2^
Pregnenolone [nM]	37.1 (31.7, 43.7)	49.2 (43.1, 56.8)	0.084	0.076	**34.1 (29.4, 39.5)**	**47.3 (43.2, 51.9)**	**0.015**	**0.119**
Pregnenolone sulphate [μM]	**4.19 (3.46, 5.02)**	**2.52 (2.13, 2.94)**	**0.01**	**0.165**	**3.7 (3.03, 4.49)**	**2.14 (1.86, 2.44)**	**0.004**	**0.157**
17-Hydroxypregnenolone [nM]	23.9 (17.6, 32.3)	39.3 (31.1, 50.1)	0.092	0.073	8.21 (5.66, 11.8)	10.8 (8.66, 13.4)	0.397	0.015
17-Hydroxypregnenolone sulphate [μM]	1.98 (1.4, 2.72)	1.8 (1.4, 2.28)	0.766	0.002	1.82 (1.35, 2.45)	1.81 (1.5, 2.18)	0.979	<0.001
16α-Hydroxypregnenolone [nM]	**15.3 (12.6, 18.7)**	**30.3 (25.3, 36.8)**	**0.002**	**0.229**	**6.92 (5.93, 8.15)**	**10.2 (9.15, 11.5)**	**0.013**	**0.125**
20α-Dihydropregnenolone [nM]	3.92 (3.35, 4.68)	5.53 (4.78, 6.54)	0.055	0.096	2.89 (2.36, 3.49)	3.97 (3.56, 4.42)	0.058	0.075
20α-Dihydropregnenolone sulphate [μM]	2.22 (1.83, 2.66)	1.8 (1.56, 2.07)	0.25	0.038	2.47 (1.98, 3.05)	1.89 (1.64, 2.17)	0.181	0.037
Dehydroepiandrosterone [nM]	8.05 (6.03, 10.9)	9.38 (7.51, 11.8)	0.589	0.008	2.56 (2.14, 3.12)	2.38 (2.14, 2.66)	0.646	0.005
DHEA sulphate [μM]	**3.28 (2.78, 3.87)**	**2.16 (1.91, 2.45)**	**0.011**	**0.166**	**3.28 (2.74, 3.87)**	**1.79 (1.55, 2.05)**	**0.001**	**0.199**
7α-Hydroxy-DHEA [nM]	0.798 (0.652, 0.968)	0.907 (0.78, 1.05)	0.494	0.013	0.458 (0.36, 0.593)	0.437 (0.378, 0.508)	0.828	0.001
7-oxo-DHEA [nM]	1.23 (0.943, 1.57)	1.78 (1.51, 2.09)	0.102	0.071	1.03 (0.783, 1.34)	1.16 (0.99, 1.37)	0.598	0.006
7β-Hydroxy-DHEA [nM]	0.259 (0.206, 0.322)	0.313 (0.265, 0.368)	0.36	0.023	0.174 (0.126, 0.247)	0.125 (0.103, 0.151)	0.241	0.029
Androstenediol [nM]	0.428 (0.328, 0.559)	0.438 (0.359, 0.536)	0.924	<0.001	**0.126 (0.101, 0.152)**	**0.178 (0.162, 0.194)**	**0.029**	**0.101**
Androstenediol sulphate [μM]	5.01 (3.84, 6.4)	4.17 (3.41, 5.04)	0.456	0.015	5.62 (4.29, 7.17)	4.67 (3.91, 5.51)	0.438	0.013
5-Androstene-3β,7α,17β-triol [pM]	42.1 (24.8, 64.5)	71.8 (54.1, 92.4)	0.176	0.049	6.59 (3.71, 11.2)	11 (8.08, 15)	0.268	0.026
5-Androstene-3β,7β,17β-triol [pM]	8.76 (4.36, 16.6)	24.8 (15.9, 38.6)	0.084	0.079	2.12 (1.23, 3.63)	3.57 (2.58, 5.03)	0.273	0.026
5-Androstene-3β,16α,17β-triol [nM]	2.58 (1.83, 3.6)	3.1 (2.41, 3.98)	0.559	0.009	2.74 (2.12, 3.59)	3.04 (2.59, 3.58)	0.659	0.004
5-Androstene-3β,16α,17β-triol sulphate [nM]	518 (435, 621)	557 (493, 631)	0.665	0.005	547 (453, 659)	509 (455, 569)	0.669	0.004
Progesterone [μM]	0.991 (0.719, 1.36)	1.44 (1.14, 1.82)	0.216	0.04	**1.76 (1.4, 2.22)**	**2.74 (2.38, 3.15)**	**0.035**	**0.094**
17-Hydroxyprogesterone [nM]	50.1 (39.3, 63.4)	66.3 (55.7, 78.9)	0.213	0.041	73.8 (58.9, 89.6)	88.8 (79, 99)	0.286	0.025
17,20α-Dihydroxy-4-pregnene-3-one [nM]	8.82 (7.13, 10.9)	13 (11.1, 15.3)	0.061	0.091	11.8 (9.25, 15.1)	15.1 (13, 17.6)	0.267	0.026
17,20α-Dihydroxy-4-pregnene-3-one, conjugated [nM]	17.4 (14.3, 21.5)	20.9 (17.8, 24.7)	0.359	0.022	20.1 (16.5, 24.6)	22.3 (19.7, 25.3)	0.552	0.007
16α-Hydroxyprogesterone [nM]	63 (47.7, 83.5)	97.5 (78.5, 122)	0.112	0.065	96.3 (75.5, 123)	135 (116, 158)	0.129	0.048
20α-Dihydroprogesterone [nM]	**67.9 (54.9, 83.5)**	**110 (94.1, 128)**	**0.018**	**0.139**	**46.4 (37.6, 56.6)**	**69.1 (61.8, 77)**	**0.022**	**0.106**
20α-Dihydroprogesterone, conjugated [nM]	62.4 (50.5, 77.9)	75.1 (63.9, 88.9)	0.374	0.021	69.8 (56.5, 85)	74.3 (65.8, 83.6)	0.722	0.003
Androstenedione [nM]	2.42 (2.01, 2.92)	2.79 (2.43, 3.22)	0.423	0.018	2.45 (1.94, 3.17)	2.32 (2.02, 2.69)	0.796	0.001
Testosterone [pM]	223 (99.6, 491)	297 (168, 524)	0.699	0.004	52.4 (29.5, 94.6)	60.1 (42.6, 85.7)	0.787	0.002
Testosterone, conjugated [nM]	34.2 (27.7, 42.9)	34.8 (30, 40.7)	0.933	<0.001	41 (31.9, 51.6)	39.1 (33.5, 45.3)	0.829	<0.001
16α-Hydroxytestosterone [nM]	8.96 (6.93, 11.5)	10.9 (9.08, 13)	0.407	0.019	8.31 (6.24, 10.9)	9.79 (8.3, 11.5)	0.497	0.01
16α-Hydroxytestosterone, conjugated [nM]	11.1 (8.54, 14.3)	11.5 (9.45, 13.8)	0.9	<0.001	10.1 (7.65, 13.2)	11.3 (9.53, 13.3)	0.656	0.004
Epitestosterone, conjugated [nM]	360 (299, 435)	362 (317, 416)	0.969	<0.001	429 (357, 515)	372 (331, 417)	0.384	0.016
5α-Dihydrotestosterone [pM]	58.2 (32.2, 103)	61.6 (40.4, 92.9)	0.916	<0.001	32.5 (17.9, 56.2)	42.1 (29.8, 58.8)	0.602	0.006
Estrone [nM]	16.1 (10, 25.2)	16 (11.4, 22.2)	0.994	<0.001	32 (22, 43.8)	49 (41.1, 57.5)	0.121	0.049
Estrone sulphate [nM]	90.9 (66.3, 127)	79.8 (63.2, 102)	0.667	0.005	86.4 (60.5, 124)	83.5 (66.9, 104)	0.913	<0.001
Estradiol [nM]	12.7 (11.4, 14.3)	14.2 (12.8, 15.9)	0.356	0.028	**18.6 (17.2, 20.1)**	**23.8 (22.5, 25.2)**	**0.002**	**0.228**
Estradiol sulphate [nM]	9.38 (7.34, 11.9)	12.5 (10.6, 14.7)	0.192	0.047	11.7 (9.41, 14.4)	10.3 (8.99, 11.8)	0.52	0.009
Estriol [nM]	161 (96.9, 266)	230 (158, 334)	0.455	0.015	374 (283, 503)	383 (323, 457)	0.927	<0.001
Estriol sulphate [μM]	3.57 (3.01, 4.27)	3.49 (3.08, 3.97)	0.888	<0.001	3.64 (2.97, 4.37)	2.89 (2.51, 3.3)	0.216	0.032
5α-Dihydroprogesterone [nM]	**124 (109, 142)**	**229 (198, 269)**	**<0.001**	**0.309**	**115 (98.5, 135)**	**289 (254, 334)**	**<0.001**	**0.415**
Allopregnanolone [nM]	**17 (14.5, 19.9)**	**31.1 (27.3, 35.5)**	**<0.001**	**0.29**	**14.4 (11.8, 17.4)**	**30.7 (27.5, 34.4)**	**<0.001**	**0.315**
Allopregnanolone sulphate [nM]	346 (264, 453)	503 (412, 616)	0.146	0.056	411 (314, 542)	552 (464, 659)	0.237	0.029
Isopregnanolone [nM]	**23.9 (20.3, 28.1)**	**56.2 (49.3, 64.2)**	**<0.001**	**0.444**	**17.3 (14, 21.2)**	**38.3 (34.7, 42.2)**	**<0.001**	**0.356**
Isopregnanolone sulphate [nM]	374 (300, 471)	426 (363, 503)	0.533	0.011	385 (310, 483)	385 (338, 441)	1	<0.001
5β-Dihydroprogesterone [nM]	27.5 (23.6, 32.8)	29.1 (25.9, 33.2)	0.719	0.004	**26.1 (21.9, 30.9)**	**35.4 (32.2, 38.9)**	**0.04**	**0.087**
Pregnanolone [nM]	24.5 (18.4, 32.3)	26.2 (21.2, 32.1)	0.806	0.002	**9.89 (7.87, 12.2)**	**15.1 (13.5, 16.9)**	**0.022**	**0.109**
Pregnanolone, conjugated [nM]	289 (226, 366)	330 (277, 390)	0.556	0.009	344 (267, 433)	374 (322, 430)	0.689	0.003
Epipregnanolone [nM]	1.85 (1.49, 2.3)	2.04 (1.73, 2.39)	0.643	0.006	1.18 (0.966, 1.43)	1.49 (1.33, 1.66)	0.171	0.04
Epipregnanolone, conjugated [nM]	61.7 (50.9, 75.5)	78 (67.2, 91.4)	0.221	0.04	66.6 (55.6, 80.2)	74.1 (66.1, 83.3)	0.518	0.009
5α,20α-Tetrahydroprogesterone [nM]	**49 (41.2, 58.5)**	**103 (88.2, 122)**	**<0.001**	**0.313**	**39.7 (31.1, 48.6)**	**79.1 (72.7, 85.7)**	**<0.001**	**0.326**
5α,20α-Tetrahydroprogesterone, conjugated [nM]	221 (153, 322)	238 (182, 312)	0.837	0.001	277 (190, 404)	249 (198, 314)	0.754	0.002
5α-Pregnane-3α,20α-diol [nM]	**6.31 (4.92, 8.04)**	**15.3 (12.8, 18.5)**	**<0.001**	**0.286**	**4.02 (2.96, 5.28)**	**10.7 (9.48, 12.1)**	**<0.001**	**0.333**
5α-Pregnane-3α,20α-diol, conjugated [μM]	3.85 (2.77, 5.32)	4.82 (3.81, 6.1)	0.461	0.015	5.02 (3.56, 7.03)	5.55 (4.49, 6.83)	0.74	0.002
5α-Pregnane-3β,20α-diol [nM]	**4.2 (3.59, 4.91)**	**9.23 (8.09, 10.6)**	**<0.001**	**0.407**	**3.75 (3.11, 4.52)**	**7.72 (6.84, 8.74)**	**<0.001**	**0.282**
5α-Pregnane-3β,20α-diol, conjugated [μM]	**2.67 (1.97, 3.56)**	**4.63 (3.81, 5.6)**	**0.04**	**0.109**	3.11 (2.34, 4.07)	4.51 (3.84, 5.27)	0.122	0.049
5β,20α-Tetrahydroprogesterone [nM]	21.9 (18, 26.9)	26.5 (22.7, 31.2)	0.332	0.025	19.4 (15.9, 23.6)	27.1 (24.2, 30.3)	0.054	0.077
5β,20α-Tetrahydroprogesterone, conjugated [nM]	130 (91.7, 182)	88.7 (67.9, 115)	0.246	0.036	174 (127, 235)	98.6 (79, 122)	0.055	0.074
5β-Pregnane-3α,20α-diol [nM]	24.7 (16.5, 35.7)	30.5 (23, 40)	0.55	0.009	6.6 (5.07, 8.63)	8.16 (6.95, 9.62)	0.368	0.017
5β-Pregnane-3α,20α-diol, conjugated [μM]	3.02 (2.27, 3.95)	2.72 (2.2, 3.32)	0.685	0.005	3.3 (2.54, 4.23)	2.8 (2.38, 3.27)	0.474	0.011
5β-Pregnane-3β,20α-diol [nM]	0.589 (0.446, 0.78)	0.818 (0.662, 1.02)	0.222	0.039	0.492 (0.396, 0.616)	0.609 (0.532, 0.7)	0.28	0.025
5β-Pregnane-3β,20α-diol, conjugated [nM]	573 (453, 731)	462 (404, 531)	0.302	0.023	438 (347, 546)	448 (381, 522)	0.916	<0.001
17-Hydroxyallopregnanolone [nM]	**0.55 (0.414, 0.711)**	**1.11 (0.958, 1.29)**	**0.003**	**0.221**	**0.35 (0.275, 0.441)**	**0.767 (0.67, 0.878)**	**<0.001**	**0.247**
17-Hydroxyallopregnanolone sulphate [nM]	**7.57 (6.28, 9.25)**	**11.5 (9.71, 14)**	**0.042**	**0.107**	**7.04 (5.73, 8.7)**	**13.1 (11.3, 15.3)**	**0.003**	**0.175**
17-Hydroxypregnanolone [nM]	1.07 (0.865, 1.32)	1.34 (1.15, 1.57)	0.258	0.035	**0.657 (0.57, 0.756)**	**0.895 (0.823, 0.973)**	**0.016**	**0.126**
17-Hydroxypregnanolone, conjugate [nM]	47.3 (38, 58.4)	62.2 (53.1, 72.7)	0.176	0.048	49.6 (39.5, 62)	64.1 (56, 73.4)	0.2	0.034
5α-Pregnane-3α,17,20α-triol [pM]	**46.5 (38.6, 56.1)**	**93 (79.8, 109)**	**<0.001**	**0.276**	**42.2 (33.1, 53.6)**	**92.1 (78.6, 108)**	**<0.001**	**0.214**
5α-Pregnane-3α,17,20α-triol, conjugated [nM]	21.7 (16.5, 28.9)	29.1 (23.4, 36.7)	0.285	0.03	26.5 (19.8, 35.9)	29.7 (24.7, 35.8)	0.675	0.004
5β-Pregnane-3α,17,20α-triol [nM]	**1.59 (1.39, 1.84)**	**2.14 (1.93, 2.38)**	**0.032**	**0.119**	1.68 (1.42, 1.98)	2 (1.81, 2.2)	0.232	0.031
5β-Pregnane-3α,17,20α-triol, conjugated [nM]	387 (305, 502)	454 (373, 560)	0.518	0.011	417 (299, 595)	465 (377, 581)	0.724	0.003
5α-Androstane-3,17-dione [nM]	0.399 (0.318, 0.51)	0.364 (0.308, 0.434)	0.674	0.005	0.186 (0.151, 0.233)	0.186 (0.164, 0.213)	0.998	<0.001
Androsterone [nM]	0.228 (0.181, 0.293)	0.15 (0.129, 0.175)	0.055	0.096	138 (110, 178)	94.3 (83.8, 106)	0.055	0.078
Androsterone sulphate [nM]	65.3 (45.5, 93.9)	58.8 (45.2, 76.4)	0.757	0.003	78.6 (55, 112)	61.2 (49, 76.3)	0.433	0.013
Epiandrosterone [nM]	0.2 (0.142, 0.284)	0.251 (0.196, 0.324)	0.49	0.013	50.3 (36.2, 69)	65.5 (54.7, 78.3)	0.341	0.02
Epiandrosterone sulphate [nM]	41.6 (35.9, 47.9)	31.9 (28.3, 35.8)	0.066	0.089	**40.2 (34.3, 46.5)**	**28.8 (25.6, 32.1)**	**0.029**	**0.096**
Etiocholanolone [pM]	41.3 (33.7, 49.9)	42.2 (36.6, 48.4)	0.901	<0.001	49.6 (41, 60.4)	40.2 (35.8, 45.1)	0.211	0.034
Etiocholanolone sulphate [nM]	9.99 (7.83, 12.8)	7.26 (6.14, 8.6)	0.156	0.055	11.4 (8.49, 15.1)	8.2 (6.73, 9.88)	0.216	0.032
Epietiocholanolone sulphate [nM]	1.96 (0.877, 4.07)	4.73 (2.81, 7.87)	0.199	0.044	3.53 (1.66, 6.77)	5.44 (3.61, 7.99)	0.466	0.011
5α-Androstane-3α,17β-diol [pM]	**11.7 (8.44, 16)**	**27 (22, 33.1)**	**0.005**	**0.198**	12.2 (9.01, 17.3)	8.65 (7.39, 10.2)	0.197	0.037
5α-Androstane-3α,17β-diol,conjugated [nM]	**13.7 (11.3, 16.6)**	**21.3 (18.3, 25.1)**	**0.023**	**0.133**	**15.4 (13.1, 18.3)**	**22.3 (19.9, 25.2)**	**0.023**	**0.103**
5α-Androstane-3β,17β-diol,conjugated [nM]	**4.52 (3.71, 5.52)**	**7.83 (6.7, 9.23)**	**0.007**	**0.182**	**4.52 (3.65, 5.57)**	**9.16 (8.08, 10.4)**	**<0.001**	**0.242**
5β-Androstane-3α,17β-diol,conjugated [nM]	4.38 (3.71, 5.3)	3.75 (3.37, 4.2)	0.318	0.028	4.59 (3.76, 5.51)	4.72 (4.19, 5.29)	0.869	<0.001
5β-Androstane-3β,17β-diol,conjugated [nM]	**0.259 (0.212, 0.319)**	**0.439 (0.376, 0.518)**	**0.011**	**0.172**	0.349 (0.264, 0.437)	0.463 (0.406, 0.521)	0.155	0.043
Cortisol [nM]	137 (115, 163)	130 (117, 145)	0.744	0.002	125 (109, 144)	104 (95, 113)	0.133	0.047
Cortisol [nM]	81.2 (67.8, 96.5)	81.4 (71.9, 91.8)	0.99	<0.001	71.5 (59.8, 85.2)	52.7 (47.1, 58.8)	0.06	0.073
Cortisone [nM]	178 (154, 206)	156 (141, 173)	0.345	0.024	196 (170, 228)	163 (149, 178)	0.147	0.043
Corticosterone [nM]	**5.84 (4.33, 7.9)**	**12 (9.55, 15.4)**	**0.017**	**0.144**	4.07 (2.95, 5.37)	5.3 (4.51, 6.17)	0.283	0.025
11-Deoxycortisol [nM]	16.9 (11.7, 24)	17 (13.1, 21.9)	0.979	<0.001	24.5 (17, 35.4)	18.2 (14.5, 22.8)	0.36	0.018
21-Deoxycortisol [pM]	**138 (92.1, 197)**	**338 (271, 419)**	**0.006**	**0.185**	206 (138, 301)	172 (134, 219)	0.607	0.006
11-Deoxycorticosterone [nM]	3.57 (2.33, 5.3)	7.08 (5.43, 9.14)	0.063	0.091	4.49 (3.45, 5.92)	7.32 (6.17, 8.74)	0.051	0.082
11-Deoxycorticosterone sulphate [nM]	100 (72.6, 141)	81.5 (63.9, 105)	0.505	0.012	117 (80.1, 169)	99.4 (78.5, 125)	0.627	0.005
11β-Hydroxyandrostenedione [nM]	6.6 (5.2, 8.38)	8.99 (7.57, 10.7)	0.172	0.05	5.31 (4.15, 6.81)	5.62 (4.85, 6.53)	0.791	0.002
11β-Hydroxytestosterone [nM]	10.1 (7.94, 12.9)	12.8 (10.8, 15.3)	0.295	0.03	9.72 (7.41, 12.8)	11.9 (10.1, 14.2)	0.398	0.015
11β-Hydroxytestosterone, conjugated, A [nM]	**60.6 (43.9, 82.2)**	**25.5 (19.7, 32.4)**	**0.007**	**0.242**	55.2 (37.9, 78.7)	32 (25.6, 39.5)	0.093	0.076
11β-Hydroxyandrosterone [pM]	55.3 (38.2, 79.1)	96.7 (74.9, 125)	0.101	0.071	36 (25.5, 49.8)	53 (43.7, 64)	0.178	0.038
11β-Hydroxyandrosterone sulphate [nM]	7.24 (5.86, 9.15)	7.01 (5.99, 8.32)	0.881	<0.001	7.78 (6.4, 9.6)	8 (7.05, 9.13)	0.878	<0.001
11β-Hydroxyepiandrosterone [pM]	25.6 (16, 40.4)	51.7 (37.2, 72.1)	0.107	0.069	**17.5 (9.92, 29.8)**	**49.7 (36.2, 68.3)**	**0.03**	**0.095**
11β-Hydroxyepiandrosterone sulphate [nM]	5.64 (3.52, 8.29)	11.2 (8.83, 13.8)	0.05	0.109	5.38 (3.41, 7.93)	9.39 (7.65, 11.4)	0.097	0.061
11β-Hydroxyetiocholanolone [pM]	82.5 (56.4, 119)	53.7 (40.1, 71)	0.236	0.038	77.7 (50.6, 116)	49.8 (37.8, 64.7)	0.241	0.029
11β-Hydroxyetiocholanolone sulphate [nM]	2.83 (2.22, 3.65)	2.47 (2.06, 2.97)	0.559	0.009	2.93 (2.27, 3.8)	2.88 (2.45, 3.39)	0.943	<0.001

The differences between twin and singleton pregnancies for each steroid were evaluated using a linear model consisting of factors Pregnancy type (Twin vs. Singleton) and Gender (Male vs. Female) adjusted for maternal age and gestational age at labour. Significant differences (*p* < 0.05) are in bold, *p*…*p*-value, η_p_^2^…effect size (0.01~small, 0.06~medium, >0.14~large).

**Table 2 ijms-25-01591-t002:** Steroid differences between dichorionic diamniotic twin pregnancy and singleton pregnancy in maternal venous blood.

	Pregnancy Type			
Steroid	Singleton	Twin	*p*	η_p_^2^
Pregnenolone [nM]	**6.21 (5.18, 7.42)**	**20.7 (17.1, 25.2)**	**<0.001**	**0.583**
Pregnenolone sulfate [nM]	**229 (189, 284)**	**150 (128, 178)**	**0.036**	**0.143**
17-Hydroxypregnenolone [nM]	2.13 (1.57, 2.8)	3.58 (2.8, 4.49)	0.066	0.116
17-Hydroxypregnenolone sulfate [nM]	4.73 (3.32, 6.76)	8.06 (5.7, 11.6)	0.162	0.066
16α-Hydroxypregnenolone [nM]	**1.24 (1.04, 1.48)**	**4.95 (3.99, 6.25)**	**<0.001**	**0.631**
20α-Dihydropregnenolone [nM]	**2.77 (2.37, 3.25)**	**5.88 (4.87, 7.25)**	**<0.001**	**0.37**
20α-Dihydropregnenolone sulfate [μM]	**0.845 (0.653, 1.1)**	**0.407 (0.317, 0.52)**	**0.01**	**0.208**
Dehydroepiandrosterone [nM]	4.74 (3.35, 6.75)	7.99 (5.67, 11.5)	0.167	0.065
DHEA sulfate [μM]	**0.895 (0.626, 1.33)**	**0.351 (0.26, 0.477)**	**0.012**	**0.199**
7α-Hydroxy-DHEA [nM]	0.343 (0.242, 0.472)	0.487 (0.359, 0.646)	0.297	0.037
7-oxo-DHEA [nM]	**0.636 (0.458, 0.862)**	**1.92 (1.49, 2.44)**	**<0.001**	**0.334**
7β-Hydroxy-DHEA [nM]	0.217 (0.161, 0.285)	0.358 (0.28, 0.45)	0.078	0.103
Androstenediol [nM]	0.587 (0.465, 0.758)	0.501 (0.405, 0.63)	0.519	0.015
Androstenediol sulfate [nM]	186 (140, 252)	152 (118, 198)	0.498	0.017
5-Androstene-3β,7α,17β-triol [pM]	107 (74.8, 148)	118 (84.5, 159)	0.788	0.003
5-Androstene-3β,7β,17β-triol [pM]	79.1 (49.1, 124)	153 (102, 226)	0.156	0.071
5-Androstene-3β,16α,17β-triol [nM]	**0.929 (0.749, 1.15)**	**2.07 (1.66, 2.59)**	**0.001**	**0.301**
5-Androstene-3β,16α,17β-triol sulfate [nM]	**252 (203, 322)**	**133 (112, 160)**	**0.005**	**0.242**
Progesterone [μM]	**0.468 (0.395, 0.559)**	**0.831 (0.673, 1.05)**	**0.009**	**0.222**
17-Hydroxyprogesterone [nM]	**30 (24.8, 36.4)**	**59.8 (48.6, 74.7)**	**0.003**	**0.27**
17,20α-Dihydroxy-4-pregnene-3-one [nM]	**7.33 (5.87, 9.18)**	**19.1 (15, 24.8)**	**<0.001**	**0.347**
17,20α-Dihydroxy-4-pregnene-3-one, conjugated [nM]	**6.39 (4.94, 8.09)**	**11.2 (9.15, 13.6)**	**0.021**	**0.169**
16α-Hydroxyprogesterone [nM]	**23.3 (19.9, 27.5)**	**52.6 (43.3, 65)**	**<0.001**	**0.406**
20α-Dihydroprogesterone [nM]	**86.8 (74.3, 101)**	**216 (181, 261)**	**<0.001**	**0.487**
20α-Dihydroprogesterone, conjugated [nM]	**29.8 (23.6, 37)**	**66.7 (55.6, 79.6)**	**<0.001**	**0.339**
Androstenedione [nM]	6.28 (5.22, 7.67)	9.24 (7.43, 11.8)	0.089	0.1
Testosterone [nM]	2.05 (1.58, 2.69)	2.63 (2, 3.52)	0.394	0.026
Testosterone, conjugated [nM]	10.6 (7.41, 14.5)	12.8 (9.34, 17)	0.579	0.011
16α-Hydroxytestosterone [nM]	**6.07 (4.98, 7.41)**	**14.4 (11.6, 18)**	**<0.001**	**0.36**
16α-Hydroxytestosterone, conjugated [nM]	**3.07 (2.27, 4.06)**	**6.36 (5, 7.99)**	**0.013**	**0.197**
Epitestosterone, conjugated [nM]	15.3 (11.9, 19.8)	21.3 (16.7, 27.4)	0.22	0.053
5α-Dihydrotestosterone [nM]	0.235 (0.161, 0.34)	0.366 (0.253, 0.523)	0.262	0.045
Estrone [nM]	17.6 (13.7, 22.6)	23.4 (18.4, 30)	0.28	0.04
Estrone sulfate [nM]	**253 (188, 329)**	**455 (367, 554)**	**0.029**	**0.159**
Estradiol [nM]	**71.8 (61.7, 83.6)**	**118 (103, 136)**	**0.003**	**0.272**
Estradiol sulfate [nM]	23.4 (19.2, 27.8)	29.1 (24.8, 33.6)	0.223	0.053
Estriol [nM]	60 (46.1, 76.1)	83.2 (66.8, 102)	0.187	0.059
Estriol sulfate [nM]	287 (216, 376)	275 (209, 356)	0.882	<0.001
5α-Dihydroprogesterone [nM]	**86.4 (68.3, 110)**	**206 (159, 272)**	**0.002**	**0.278**
Allopregnanolone [nM]	**33.9 (26.9, 42.4)**	**67 (54, 83.2)**	**0.006**	**0.23**
Allopregnanolone sulfate [μM]	1.78 (1.39, 2.29)	2.03 (1.58, 2.61)	0.632	0.008
Isopregnanolone [nM]	**10.2 (7.76, 13)**	**56 (45.4, 69.3)**	**<0.001**	**0.646**
Isopregnanolone sulfate [μM]	1.08 (0.806, 1.44)	2 (1.49, 2.74)	0.056	0.125
5β-Dihydroprogesterone [nM]	**1.24 (0.832, 1.77)**	**4.66 (3.54, 6.05)**	**<0.001**	**0.372**
Pregnanolone [nM]	18.6 (15.7, 22.1)	25.3 (21.5, 29.9)	0.094	0.097
Pregnanolone, conjugated [μM]	1.06 (0.842, 1.3)	1.16 (0.936, 1.42)	0.677	0.006
Epipregnanolone [nM]	**0.882 (0.709, 1.09)**	**1.62 (1.33, 1.97)**	**0.009**	**0.214**
Epipregnanolone, conjugated [nM]	**182 (143, 230)**	**357 (279, 460)**	**0.013**	**0.203**
5α,20α-Tetrahydroprogesterone [nM]	**34.8 (27.5, 43.9)**	**112 (89.1, 141)**	**<0.001**	**0.453**
5α,20α-Tetrahydroprogesterone, conjugated [nM]	160 (95.8, 260)	263 (166, 409)	0.329	0.033
5α-Pregnane-3α,20α-diol [nM]	**24.8 (19.5, 31.3)**	**88 (70.4, 111)**	**<0.001**	**0.496**
5α-Pregnane-3α,20α-diol, conjugated [μM]	12.1 (8.86, 16.6)	8.14 (6.04, 11)	0.229	0.05
5α-Pregnane-3β,20α-diol [nM]	**4.01 (3.22, 4.95)**	**21 (17.2, 25.8)**	**<0.001**	**0.682**
5α-Pregnane-3β,20α-diol, conjugated [μM]	7.66 (5.67, 10.3)	9.41 (7.06, 12.6)	0.511	0.015
5β,20α-Tetrahydroprogesterone [nM]	**1.66 (1.39, 1.97)**	**3.63 (3.07, 4.31)**	**<0.001**	**0.397**
5β,20α-Tetrahydroprogesterone, conjugated [nM]	53.3 (33.3, 83.2)	56.2 (36, 86.1)	0.909	<0.001
5β-Pregnane-3α,20α-diol [nM]	**12.4 (10.4, 14.7)**	**26.8 (22.4, 32.4)**	**<0.001**	**0.376**
5β-Pregnane-3α,20α-diol, conjugated [μM]	4.81 (3.81, 5.94)	3.26 (2.52, 4.1)	0.125	0.082
5β-Pregnane-3β,20α-diol [nM]	**0.463 (0.326, 0.653)**	**1.18 (0.843, 1.68)**	**0.014**	**0.19**
5β-Pregnane-3β,20α-diol, conjugated [μM]	1.07 (0.839, 1.35)	0.742 (0.565, 0.95)	0.167	0.065
17-Hydroxyallopregnanolone [nM]	**0.386 (0.286, 0.51)**	**1.59 (1.26, 2)**	**<0.001**	**0.51**
17-Hydroxyallopregnanolone sulfate [nM]	**15.1 (11.9, 19.1)**	**49 (37.5, 65.3)**	**<0.001**	**0.423**
17-Hydroxypregnanolone [nM]	**1.01 (0.861, 1.18)**	**1.51 (1.3, 1.77)**	**0.019**	**0.177**
17-Hydroxypregnanolone, conjugate [nM]	74.4 (61.4, 89.9)	111 (92.6, 133)	0.05	0.126
5α-Pregnane-3α,17,20α-triol [pM]	**100 (76.3, 131)**	**282 (214, 377)**	**0.001**	**0.31**
5α-Pregnane-3α,17,20α-triol, conjugated [nM]	47.6 (34.5, 63.9)	70.8 (53.8, 91.9)	0.196	0.057
5β-Pregnane-3α,17,20α-triol [nM]	4.63 (3.83, 5.57)	6.54 (5.51, 7.74)	0.077	0.104
5β-Pregnane-3α,17,20α-triol, conjugated [nM]	380 (301, 485)	441 (367, 535)	0.518	0.011
5α-Androstane-3,17-dione [nM]	**0.379 (0.306, 0.471)**	**0.635 (0.509, 0.799)**	**0.035**	**0.149**
Androsterone [nM]	0.461 (0.379, 0.566)	0.661 (0.538, 0.828)	0.109	0.086
Androsterone sulfate [nM]	291 (220, 381)	188 (140, 249)	0.15	0.073
Epiandrosterone [nM]	**0.143 (0.107, 0.19)**	**0.553 (0.414, 0.747)**	**<0.001**	**0.417**
Epiandrosterone sulfate [nM]	71.9 (57, 91.3)	63 (51, 78.2)	0.58	0.011
Etiocholanolone [pM]	148 (125, 175)	164 (140, 193)	0.55	0.013
Etiocholanolone sulfate [nM]	22.4 (17.3, 30)	18.3 (14.7, 23.4)	0.451	0.02
Epietiocholanolone sulfate [nM]	3.96 (2.53, 6.22)	3.4 (2.21, 5.24)	0.748	0.004
5α-Androstane-3α,17β-diol [pM]	66.8 (53.7, 84.5)	86.2 (68.3, 112)	0.312	0.037
5α-Androstane-3α,17β-diol, conjugated [nM]	**13.7 (11.3, 16.6)**	**21.3 (18.3, 25.1)**	**0.023**	**0.133**
5α-Androstane-3β,17β-diol, conjugated [nM]	**9.58 (7.44, 12.1)**	**18.8 (15.2, 23.1)**	**0.008**	**0.218**
5β-Androstane-3α,17β-diol, conjugated [nM]	2.46 (1.99, 3.07)	3.13 (2.53, 3.93)	0.302	0.037
5β-Androstane-3β,17β-diol, conjugated [nM]	0.34 (0.24, 0.456)	0.532 (0.415, 0.663)	0.129	0.08
Cortisol [nM]	**808 (686, 970)**	**553 (486, 632)**	**0.023**	**0.16**
Cortisol [nM]	**773 (678, 876)**	**579 (505, 660)**	**0.045**	**0.131**
Cortisone [nM]	177 (157, 200)	185 (165, 208)	0.713	0.005
Corticosterone [nM]	24.8 (19.4, 32)	30.6 (23.7, 39.9)	0.441	0.021
11-Deoxycortisol [nM]	8.78 (5.72, 13.3)	18.2 (12.3, 26.9)	0.097	0.092
21-Deoxycortisol [pM]	100 (64.6, 150)	131 (87.9, 190)	0.535	0.013
11-Deoxycorticosterone [nM]	**0.579 (0.307, 0.939)**	**7.44 (5.72, 9.6)**	**<0.001**	**0.654**
11-Deoxycorticosterone sulfate [nM]	**3.34 (1.8, 5.72)**	**15.6 (10.3, 23.1)**	**0.005**	**0.246**
11β-Hydroxyandrostenedione [nM]	46.7 (37.5, 59.2)	58 (45.7, 75.5)	0.391	0.026
11β-Hydroxytestosterone [nM]	**7.49 (6.4, 8.84)**	**15 (12.2, 19)**	**0.001**	**0.319**
11β-Hydroxytestosterone, conjugated, A [nM]	8.78 (5.27, 14.2)	14.8 (9.79, 22.1)	0.266	0.059
11β-Hydroxyandrosterone [pM]	**162 (131, 201)**	**376 (305, 465)**	**<0.001**	**0.348**
11β-Hydroxyandrosterone sulfate [nM]	8.54 (6.71, 11)	8.76 (6.94, 11.1)	0.923	<0.001
11β-Hydroxyepiandrosterone [pM]	**16.3 (11.7, 22.5)**	**89 (63, 128)**	**<0.001**	**0.462**
11β-Hydroxyepiandrosterone sulfate [nM]	7.13 (4.77, 10.5)	9.52 (6.68, 13.6)	0.473	0.02
11β-Hydroxyetiocholanolone [pM]	232 (166, 332)	128 (93.8, 176)	0.094	0.101
11β-Hydroxyetiocholanolone sulfate [nM]	1.5 (1.04, 2.16)	1.14 (0.798, 1.62)	0.479	0.018

The differences between twin and singleton pregnancies for each steroid were evaluated using a linear model consisting of factors Pregnancy type (Twin vs. Singleton) and Gender (Male vs. Female) adjusted for maternal age and gestational age at labour. Significant differences (*p* < 0.05) are in bold, *p*…*p*-value, η_p_^2^…effect size (0.01~small, 0.06~medium, >0.14~large).

**Table 3 ijms-25-01591-t003:** Characteristics of studied groups, included patients only.

	Pregnancy Type	
Characteristic	Singleton (Control)	Biamniotic Bichorionic Twin
Number of included patients	19	24
Race	Caucasian (19)	Caucasian (24)
Mean age (at the time of delivery)	36.9 (22, 45)	32 (27, 44)
Parity	0.79 (0, 1)	0.71 (0, 4)
Spontaneous conception	15	17
Conception after IVF methods	4	7
Mean weight before pregnancy (kg)	66.6 (49, 122)	73.9 (53, 112)
Mean weight at the time of delivery (kg)	80.1 (65, 118)	94.2 (70, 117)
Mean height (cm)	165.9 (145, 180)	164 (155, 186)
Gestational age at the time of delivery	38.34 (38 + 1, 41 + 1)	37.99 (35 + 3, 39 + 3)
Newborn male sex	9	30
Newborn female sex	10	18
Mean newborn weight	3391.6 (2780, 4050)	2775.5 (1780, 3570)
Mean newborn length *	49.8 (47, 51)	48.1 (43, 51)

Main characteristics of the study and control group. * Mean newborn length in twin foetuses is measured in head presentation only. For this reason, the table does not contain born in breech presentation, as length is measured several days after delivery.

## Data Availability

The data presented in this study are available on request from the corresponding author. The data are not publicly available due to privacy restrictions.

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
