# Peer review of "Steroid Metabolome Analysis in Dichorionic Diamniotic Twin Pregnancy"

_ijms, 2024, doi:10.3390/ijms25031591_

Round 1
Reviewer 1 Report
Comments and Suggestions for Authors
This paper presents a survey of steroids produced during pregnancy by pregnant women (maternal blood and cord venous) and by fetuses (cord arterial blood). As the reviewer, I need help finding the real aim of this study. What hypotheses did the authors put forward? What changes were the authors expected to find between the studied groups? The authors included healthy twin dichorionic pregnancies and single pregnancies.
1. The authors should present the characteristics of the studied groups, especially maternal age, pregnancy (BMI), weight gain, parity, and fetal mass.
2. The main weak point is the difference in delivery time in twin pregnancies (34–39 weeks of pregnancy), which may cause differences in steroid metabolome concentration between studied patients. This period is also difficult to compare with singleton pregnancies, which delivered between 38 and 41 weeks of pregnancy.
3. I suggest showing the alphabetical list of studied steroids and, in addition, any flow chart of correlations between studied metabolites to show the readers what the authors have found.
Author Response
Thank you very much for taking the time to review this manuscript. Please find the detailed responses attached and the corresponding revisions/corrections highlighted in in the re-submitted files.

Reviewer 2 Report
Comments and Suggestions for Authors
Journal: IJMS Manuscript ID: ijms-2773599
Authors: Andrej Černý et al.
Title: “Steroid metabolome analysis in dichorionic diamniotic twin pregnancy”
The investigators of the current study explored the biosynthesis, transportation, and impact of steroids in instances of multiple pregnancies. The research focused on comparing women with multiple pregnancies to those with singleton pregnancies in order to characterize the steroid metabolome in twins, compare metabolome profiles between multiple and singleton pregnancies, and examine gender-based variations in fetal steroid metabolism. The following points should be considered:
Comments:
- The abstract should include more information regarding the study's results while also providing the main conclusion(s) of the paper. If necessary, you can consider providing a more concise description of the study's background and introduction to reduce its length.
- In terms of clarity and reproducibility, it is important that the authors provide all the exclusion and inclusion criteria. Additionally, did the authors assess whether the pregnant women had a history of PCOS, insulin resistance, whether this was their first pregnancy, and whether the pregnancies were naturally conceived or IVF-assisted? Was a planned caesarean section intended for all pregnancies in both the study and control groups?
- Please provide further information regarding the sampling method. For example, were the pregnant individuals selected randomly, or were all eligible participants chosen in a continuous order?
- According to the authors, “Immediately after birth, arterial and venous blood was collected separately from each twin’s umbilical cord.” Did the authors perform all the processes and procedures in the same way for both multiple and singleton pregnancies?
- Have the authors accounted for BMI, glucose levels, and insulin resistance in their analysis?
- Please provide a table that presents the demographic and essential characteristics of the participants in each study group. Additionally, please assess whether there were any significant differences observed between these groups.
- In accordance with the authors' statement, "Our previous research showed that samples taken and frozen using this technique are very stable, and the concentrations of steroids do not change." Please provide the relevant reference as appropriate.
- Could you provide additional information pertaining to the clinical data in section #2.4?
- Please elaborate on the statistical analysis conducted, including the specific tests employed and the statistical software uzed. Did the authors perform a power calculation?
- Please specify the type of specimens in the relevant section of the Methods.
- Did the authors consider performing a pathway analysis?
- It would be beneficial to include a section in the discussion addressing any limitations of the study.
- Please incorporate a concise paragraph summarizing the study's conclusions.
Author Response
Thank you very much for taking the time to review this manuscript. Please find the detailed responses below and the corresponding revisions/corrections highlighted in in the re-submitted files.

Round 2
Reviewer 1 Report
Comments and Suggestions for Authors
I have no more comments to this paper
Author Response
Dear Reviewer,
Thank you for time you dedicated to review this article. I appreciate this a lot and hope to collaborate in the future.
Best regards
Andrej Černý
Reviewer 2 Report
Comments and Suggestions for Authors
The authors of the present article have tried to respond to my comments and suggestions and have made the appropriate changes to the paper, thereby improving the manuscript. However, the following points still need to be considered:
It would be beneficial to add the relevant information regarding the power calculation performed for the study.
Moreover, regarding comment #5 about BMI, and given the association between adipose tissue and steroid hormones, did the authors consider the impact of BMI in their findings? It would be important for the authors to include this variable in the multivariate regression models.
In Table 1, please indicate whether a statistical difference was observed between the two groups.
Author Response
Dear Reviewer,
Thank you very much for taking the time to review notes and for your further comments. Please find responses attached.
